# A Comparison Study of the Nutrient Fluxes in a Newly Impounded Riverine Lake (Longjing Lake): Model Calculation and Sediment Incubation

Cheng Du [1,2], Yan'an Pan [1,3], Wenzhong Tang [1,4], Qiansheng Yue [2] and Hong Zhang [1,4,*]

[1] State Key Laboratory of Environmental Aquatic Chemistry, Research Center for Eco-Environmental Sciences, Chinese Academy of Sciences, Beijing 100085, China; ducheng98@outlook.com (C.D.); panyanan123@yeah.net (Y.P.); wztang@rcees.ac.cn (W.T.)
[2] College of Chemistry & Environmental Engineering, Yangtze University, Jingzhou 434023, China; yueqiansheng@163.com
[3] Lanzhou LS Heavy Equipment Co., Ltd., Lanzhou 730050, China
[4] Colleague of Resources and Environment, University of Chinese Academy of Sciences, Beijing 100049, China
* Correspondence: hongzhang@rcees.ac.cn; Tel./Fax: +86-10-6292-2031

**Abstract:** Diffusion flux is an essential tool to estimate the contribution of internal nitrogen and phosphorus in eutrophic lakes. There are mainly two methods, i.e., model calculation based on in-situ porewater sampling and water quality monitoring in laboratory incubation. The results obtained by the two methods are rarely compared, decreasing the validity of internal contribution and following management strategies. In this study, sediment samples were collected from a lake in China, then the fluxes were estimated by model calculation and laboratory incubation. The results show that there is an order of magnitude difference in the fluxes measured by these two methods. The mean values of ammonia ($NH_4^+$-N) and soluble reactive phosphate (SRP) obtained from the model calculations were 24.4 and 1.30, respectively. The mean values of $NH_4^+$-N and SRP obtained in the undisturbed group of sediment incubation were 7.84 and 5.47, respectively, and in the disturbed group of sediment incubation were 16.2 and 4.06, respectively. Sediment incubation is a combination of multiple influencing factors to obtain fluxes, while porewater model is based on molecular diffusion as the theoretical basis for obtaining fluxes. According to the different approaches of the two methods, sediment incubation is recommended as a research tool in lake autochthonous release management when the main objective is to remove pollution, while the porewater model is recommended as a research tool when the main objective is to control pollution. When assessing the diffusive flux of nitrogen, it is recommended to choose the stable form of total dissolved nitrogen to discuss the flux results.

**Keywords:** sediment incubation; nutrient fluxes; flow disturbed; model calculation

## 1. Introduction

Nutrient release from sediments plays an important role in ecosystem cycles and was a major source of substrates for primary productivity in aquatic systems [1,2]. In several shallow lakes, autochthonous nutrients contributed to a large proportion of the summer bloom outbreak [3,4], acting as the main factor controlling primary lake productivity in global eutrophication studies. While in some reservoirs, autochthonous releases were stimulated by flow in the water column when the reservoir is initially stored resulting in disruption of the environment's homeostasis and thus making it more susceptible to eutrophication [5,6]. Prediction and control of nutrient loading require knowledge of the key factors that control the autochthonous release [7]. Understanding the transport fluxes of nitrogen and phosphorus at the sediment-water interface (SWI) was a prerequisite for controlling the trophic state of the aquatic environment [8].

Flux studies were effective methods used as an assessment of the risk associated with contaminated sediments [9,10]. There were mainly two kinds of methods to determine the fluxes of compounds released from the SWI [11]. One was a model calculation based on Fick's first law, including the estimate of the interstitial water concentration diffusion model [12], flux incubation [13,14], benthic chamber [15], mass conservation model [16], and flume experiment [17]. The other one was incubation, such as static incubation [18,19]. Most of the researchers reported their results obtained by either model calculation or incubation. However, the comparison of the two methods was rarely carried out, which was important when fluxes from different sites were compared.

In our previous study on a lake in Southwest China, i.e., the Longjinghu Lake located in the center of Chongqing Garden Expo Park, the porewater fluxes of N and P were calculated as a one-dimensional transport-reaction model [20]. Here, samples of five sediments were collected and then further estimated by laboratory incubation for flux. The experimental objectives are: (1) To quantify the main nutrient sources using sediment incubations and estimate their contributions to lake nitrogen and phosphorus input, (2) to compare the results of the two studies and discuss the applicability of the two methods in the experiments. The results would validate the flux estimation for internal contribution and following management strategies.

## 2. Materials and Methods

### 2.1. Study Area

Experimental sediments were taken from Longjinghu Lake, located in the center of Chongqing Garden Expo Park. The total water surface area of Longjing Lake is about 0.67 km$^2$, and about 0.53 km$^2$ in the Chongqing Garden Park Expo, with a total reservoir capacity of 6.63 million m$^3$ and a regulating reservoir capacity of 4.25 million m$^3$. There are two input rivers, Zhaojiaxi River (catchment area of 15 km$^2$) and Longjingguo River (catchment area of 5 km$^2$), providing sufficient water to Longjinghu Lake for the management and irrigation of the park landscape. The area of Longjinghu Lake is a temperate continental climate (106°56′ N, 29°69′ W), with a water depth of about 20~30 m and a water exchange cycle of about 2.5 a. The water source is mainly recharged by rainfall runoff. The main pollutants are nitrogen, phosphorus, and TSS, and the water quality of the planned lake is surface water environmental quality standard IV.

### 2.2. Sediment Sampling

The water surface of Longjinghu lake is divided into five regions based on the depth of water, which are LH, LG, HD, ZJ, and KW, where KW and ZJ are the new submerged zones, and LH, LG, and HD are the original lakes. For the experimental samples, there were 17 sediment cores collected from Longjinghu Lake to extract porewater for calculating the diffusion flux of internal nutrients in 2013 [20]. The same sampling sites, namely LH, LG, HD, ZJ, and KW, were set for comparison with the previous study [20], as shown in Figure 1. Sediment and overlying water were collected using a self-weight column sediment sampler (Corer 60, Uwitec, Salzkammergut, Austria). When sampling was complete, the samples were immediately sealed and transported back to the laboratory and were allowed to stand for 24 h to eliminate the effects of the collection and transport process. Afterward, the overlying water was collected with a siphon tube and separated into incubation cylinders with the sediment.

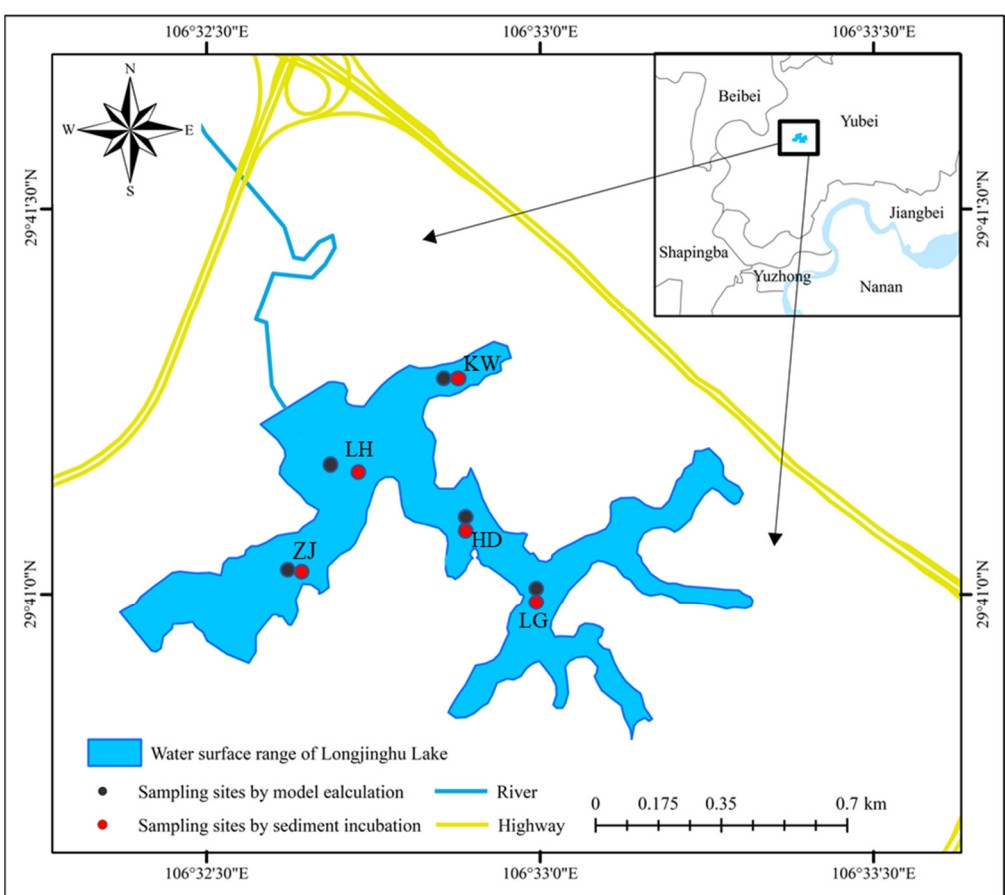

**Figure 1.** Distribution of sediment sampling sites in Longjinghu Lake.

### 2.3. Incubation and Analyses

For the incubation, the five sediment cores with overlying water were put for 24 h. The disturbed group, the undisturbed group, and the control group were set for the incubations of each sample site (LH, LG, HD, ZJ, and KW, respectively). The undisturbed group was used to investigate the internal diffusion flux statically, while the external interference conditions, such as wind and wave, were taken into account in the disturbed group, which was realized by artificial agitation five times (2 min for each time) a day. The control group was set to observe the blank for all the periods of the incubations. A certain amount of distilled water was added to the incubation cylinders regularly every day to compensate for the evaporation loss. The experimental setup design is shown in Figure S1.

Water samples (30 mL) were collected at the incubating time of 0, 0.25, 0.5, 1, 1.5, 2, 2.5, 3, 4, 5, 6, 7, 8, 9, 10, 11, 12, 13, 14, 15, 17, 19 d. All water samples were filtered through 0.45 μm pore size cellulose acetate membranes (Whatman). The concentrations of dissolved total nitrogen (DTN), nitrate ($NO_3^--N$), ammonia ($NH_4^+-N$), dissolved total phosphate (DTP), and soluble reactive phosphate (SRP) were analyzed according to the standard method [21].

### 2.4. Methodology of Flux Estimation for N and P

The flux of N and P during the incubations was calculated based on the monitoring data, then it was compared with the results obtained by a one-dimensional transport-reaction model at the same sites [20].

(1)    Flux of N and P during incubations

The formula for nutrient concentrations in the overlying water in the incubation cylinders is as follows:

$$Q_0 = C_0 V_1 \tag{1}$$

$$Q_1 = C_1 V_1 \tag{2}$$

$$Q_i = C_i V_1 + V_2(C_1 + C_2 + C_3 + \cdots + C_{(i-1)}) - V_2(D_1 + D_2 + D_3 + \cdots + D_{(i-1)}) \tag{3}$$

where i is the i-th sampling, i.e., 1, 2, $\cdots$ , 22 (the whole experiment was measured 22 times), $V_1$ is the volume of overlying water of the incubation cylinders (L), $V_2$ is the volume of sample required for each measurement (L), $C_i$ is the concentration value of overlying water measured by the sample (mg/L), and $D_i$ is the supplemental water concentration measured by different samples (mg/L).

The concentrations of nutrients in the incubations were calculated according to the law of the conservation of mass of the incubation system by the formula as follows:

$$F_i = [Q_i - Q_{(i-1)}]/(A \cdot \Delta t_i) \tag{4}$$

where, Fi is the diffusion flux obtained from each sampling (mg/(m$^2$·d)), i is the i-th sampling, i.e., 1, 2, $\cdots$, 22 (the whole experiment was measured 22 times), $Q_i$ is the nutrient mass obtained from different incubation cylinders, $Q_0$ is the nutrient mass at the beginning of the incubation (mg), A is the sediment-water interface area (m$^2$), and $\Delta t_i$ is the time of the incubation period (d).

(2)    One-dimensional transport-reaction model for porewater fluxes of N and P

A one-dimensional transport-reaction model [22] was adopted to calculate the fluxes of N and P based on the porewater profiles.

$$F = \varphi \cdot D_s \cdot [\delta_c / \delta_x] \tag{5}$$

where, $\varphi$ is the tortuosity obtained from the formula in the reference, $\delta_c / \delta_x$ is the content gradient of dissolved nutrient between the surface porewater and the overlying water, (mg/(L·cm)), $D_s$ is the sediment diffusion coefficient of species derived from the diffusion coefficient ($D_0$) of the species at infinite dilution (m$^2$/s), and their relationships could be calculated by an empirical formula: Ds = $\varphi D_0$ ($\varphi < 0.7$), $D_s = \varphi^2 D_0$ ($\varphi > 0.7$), the diffusion coefficient ($D_0$) in water is $6.12 \times 10^{-6}$ cm$^2$/s for $HPO_4^{2-}$ and $17.6 \times 10^{-6}$ cm$^2$/s for $NH_4^+$-N.

$$\varphi = \{[W_w - W_d] \cdot 100\%\} / \{[W_w - W_d] + W_d / \rho\} \tag{6}$$

where $W_w$ is the fresh sediment weight, $W_d$ is the dry sediment weight, $\rho$ is the ratio of the average density of surface sediment to water density, generally taken as 2.5. The flux results of nitrogen and phosphorus can be found in our previous study [20].

*2.5. Statistical Methods*

Data statistics were analyzed using SPSS, and significant differences between data were demonstrated by t-test.

**3. Results and Discussion**

*3.1. Flux of DTP and SRP in Incubation*

The mass accumulation of DTP and SRP in the overlying water of both the disturbed and undisturbed groups showed a steady linear increase over time (Figure 2). The trends of DTP and SRP were similar, with a slight decrease in mass in both groups by the end of incubations. For the control group, the mass accumulation was relatively steady. It could also be seen that The DTP and SRP concentrations in the overlying water collected by LH, LG, HD, and KW remained around 0.01 mg/L in the control group throughout the incubation period (Figure S2). In LH and LG, the mass accumulation of the disturbed group was lower than that of the undisturbed group. In KW, the mass accumulation of the disturbed group was higher than that of the undisturbed group. In ZJ and HD, the mass accumulation of the disturbed group was similar to that of the undisturbed group. As shown in Figure 2, the slope of the control group was close to 0, so phosphorus transformation in the overlying water column could be neglected in the incubations.

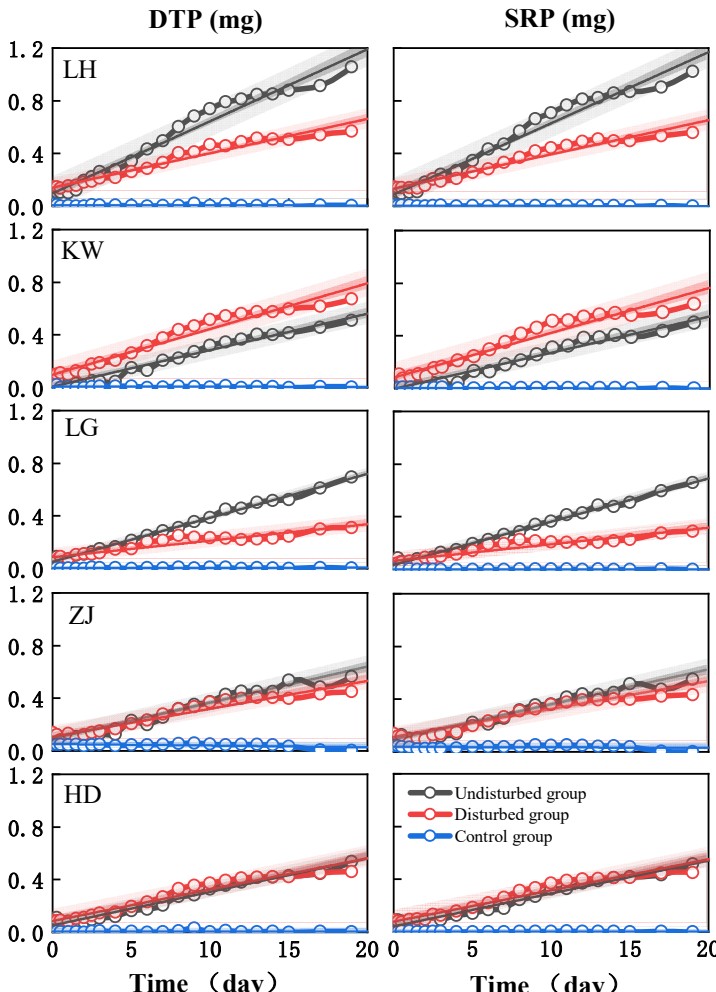

**Figure 2.** Mass changes of dissolved total phosphate (DTP) and soluble reactive phosphate (SRP) at different sampling sites over time during the experiment. LH, kW, LG, ZJ, and HD are different sampling sites, respectively. The full line represents the result of linear fitting based on the data.

The results of diffusion fluxes based on sediment incubation of DTP and SRP are depicted in Figure 3. In the disturbed group, the mean fluxes of SRP for LH, KW, LG, ZJ, and HD were 7.35, 11.5, 6.80, 4.29, and 6.05, respectively. In the undisturbed group, the mean fluxes of SRP for LH, KW, LG, ZJ, and HD were 17.9, 2.77, 11.6, 5.42, and 3.91, respectively. In the disturbed group, the mean DTP fluxes for LH, KW, LG, ZJ, and HD were 8.20, 12.5, 6.71, 6.31, and 6.41, respectively. In the undisturbed group, the mean DTP fluxes for LH, KW, LG, ZJ, and HD were 18.4, 3.43, 11.0, 5.96, and 4.75, respectively. Among them, the DTP and SRP of all sampling sites showed sediment release to the overlying water, which indicated the risk of autochthonous release from Longjing Lake.

In the comparison between the disturbed and undisturbed groups, there were significant differences in the flux results for LH, KW, and LG ($p < 0.05$) and no significant differences in the flux results for ZJ and HD ($p > 0.05$). Disturbance only showed a relatively limited effect on phosphorus release (Figure 3). In general, fluxes were higher in the disturbed group than in the undisturbed group because the disturbance increases the release rate. Among LH and LG, the flux of the undisturbed group was higher than the disturbed group, which may be related to the phosphorus morphological conversion in the anaerobic environment. With the increase in incubation time, the concentrations of both groups gradually converged to the same level except for LH (Figure S2). This indicates that in the relatively short term, the disturbed water flow had a larger effect on the accumulation of DTP and SRP loads. However, in the long term, the effect of water

flow disturbance diminished. Thus, the comparison between the different groups showed that the undisturbed group had a continuous and steady increase in DTP and SRP, and the disturbed group probably reduced the release of DTP and SRP from the sediment.

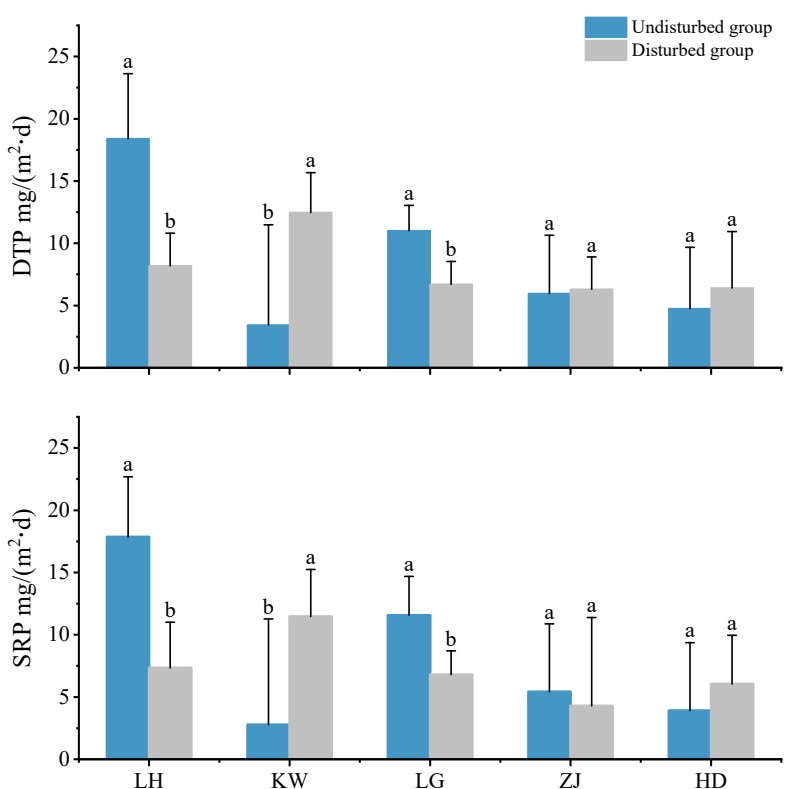

**Figure 3.** Diffusion fluxes of dissolved total phosphate (DTP), and soluble reactive phosphate (SRP) at different sampling sites during the incubations. LH, kW, LG, ZJ, and HD represent different sampling sites. The labels on the error bar represent the results of the significance test, with identical letters representing no significant difference and different letters representing significant difference.

### 3.2. Flux of DTN, $NO_3^-$-N, and $NH_4^+$-N in Incubation

The changes in mass accumulation of DTN, ammonia, and nitrate are shown in Figure 4. The ammonia accumulation showed a trend of increasing and then decreasing during the incubation period for all sites except for ZJ, whereas the nitrate accumulation showed an increasing trend after the middle of the incubation period. This indicates that the nitrogen in the water column was endowed in the form of ammonia in the early stage of incubation, and gradually converted to nitrate with the increase of incubation time. Total nitrogen showed an increasing linear trend. In LH and LG (except nitrate in LH), the mass accumulation of the disturbed group was higher than that of the undisturbed group. In ZJ, the mass accumulation of the disturbed group was lower than that of the undisturbed group. For KW and HD, the mass accumulation of the disturbed group approximated that of the undisturbed group. As shown in Figure S3, the concentration of DTN in the control group overlying water remained at approximately 0.3~1.2 mg/L in all of the five sites above. This indicated that DTN in the overlying water also changed without sediment involvement ($p < 0.05$). The mass accumulation of DTN in both undisturbed and disturbed groups reached the maximum values on day 15. The dynamic system formed by the disturbance influenced the state of the overlying water, which in turn, influenced the release of DTN.

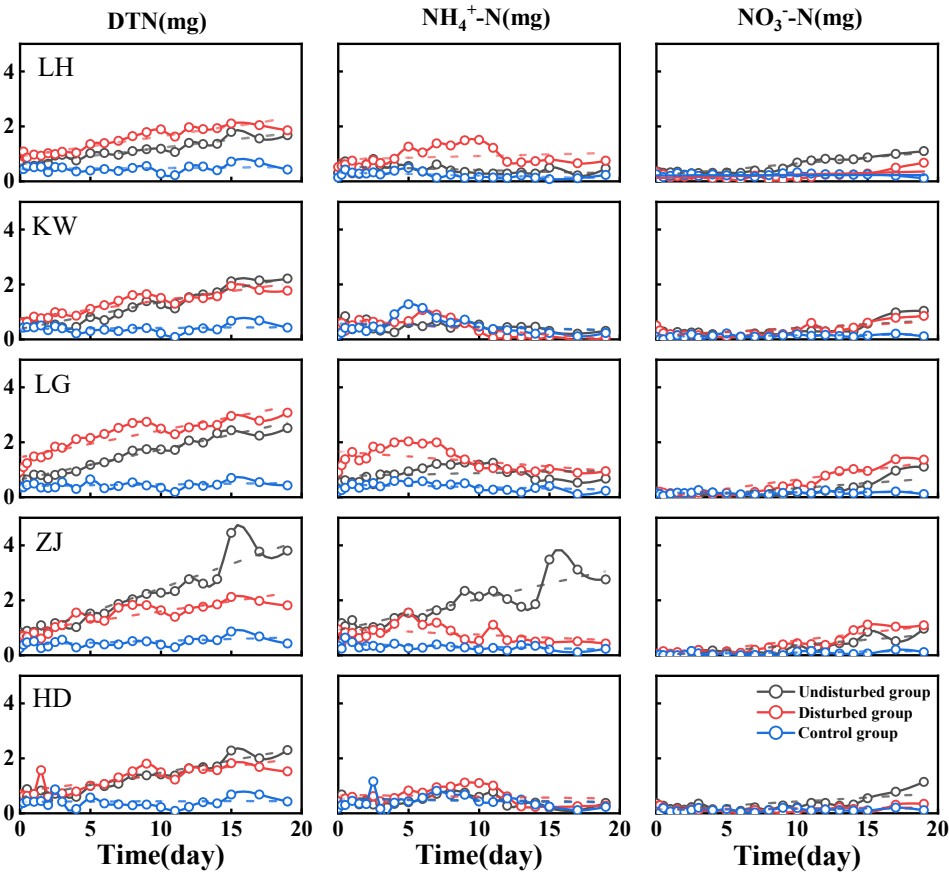

**Figure 4.** Mass changes of dissolved total nitrogen (DTN), nitrate ($NO_3^-$-N), and ammonia ($NH_4^+$-N) at different sampling sites over time during the experiment. LH, kW, LG, ZJ, and HD are different sampling sites, respectively. The dotted line represents the result of linear fitting based on the data.

Figure 5 shows the results of diffusion fluxes based on sediment incubation of DTN, $NH_4^+$-N, and $NO_3^-$-N. In the disturbed group, the $NH_4^+$-N fluxes for LH, KW, LG, ZJ, and HD were 25.0, −6.86, 69.0, 1.14, and −7.10, respectively. In the undisturbed group, the $NH_4^+$-N fluxes for LH, KW, LG, ZJ, and HD were 8.46, −12.0, 13.9, 46.1, and −17.2, respectively. In the disturbed group, the $NO_3^-$-N fluxes for LH, KW, LG, ZJ, and HD were −13.0, −29.0, −24.5, −44.82, and −44.3, respectively. In the undisturbed group, the $NO_3^-$-N fluxes for LH, KW, LG, ZJ, and HD were 2.92, −27.4, −35.6, 49.8, and −34.8, respectively. In the disturbed group, the DTN fluxes for LH, KW, LG, ZJ, and HD were 44.6, 33.1, 98.0, 27.4, and 18.7, respectively. In the undisturbed group, the DTN fluxes for LH, KW, LG, ZJ, and HD were 14.8, 12.4, 39.1, 45.4, and 16.3, respectively.

As shown in Figure 5, the DTN flux in the disturbed group was slightly higher than that in the undisturbed group (except ZJ). The fluxes of $NH_4^+$-N in LH, LG, and ZJ are above 0, KW and HD are below 0. The fluxes of $NO_3^-$-N are below 0 except for LH, and no significant difference was found in the significance test, which indicates that the disturbance did not affect the conversion of nitrate. Nitrogen was active in the aquatic environment [23], with active biochemical reactions of transport conversion in the water (e.g., anaerobic ammonium oxidation, anammox), where anaerobic $NH_4^+$-N was found to be the main pathway of nitrogen loss, showing a dynamic equilibrium in the water [24]. DTN and $NO_3^-$-N had the slight potential to release other organisms or nutrients, while $NH_4^+$-N could be absorbed or transformed by aquatic organisms [25,26]. The disturbance might promote the conversion of $NH_4^+$-N to $NO_3^-$-N in the water, leading to a significant reduction in $NH_4^+$-N [27]. An order of magnitude difference was found between the load accumulation of ammonia in the disturbed group and the undisturbed group. This could probably be attributed to the disrupting of the dynamic equilibrium of ammonia in the

water, which, in turn, led to a large increase or decrease of nitrogen in the water [28]. Due to the instability of ammonia and nitrate in the water column, it is recommended to evaluate the dynamic processes of nitrogen in the form of total nitrogen rather than any form of nitrogen $NO_3^-$-N and $NH_4^+$-N) alone when discussing nutrient loading to lakes.

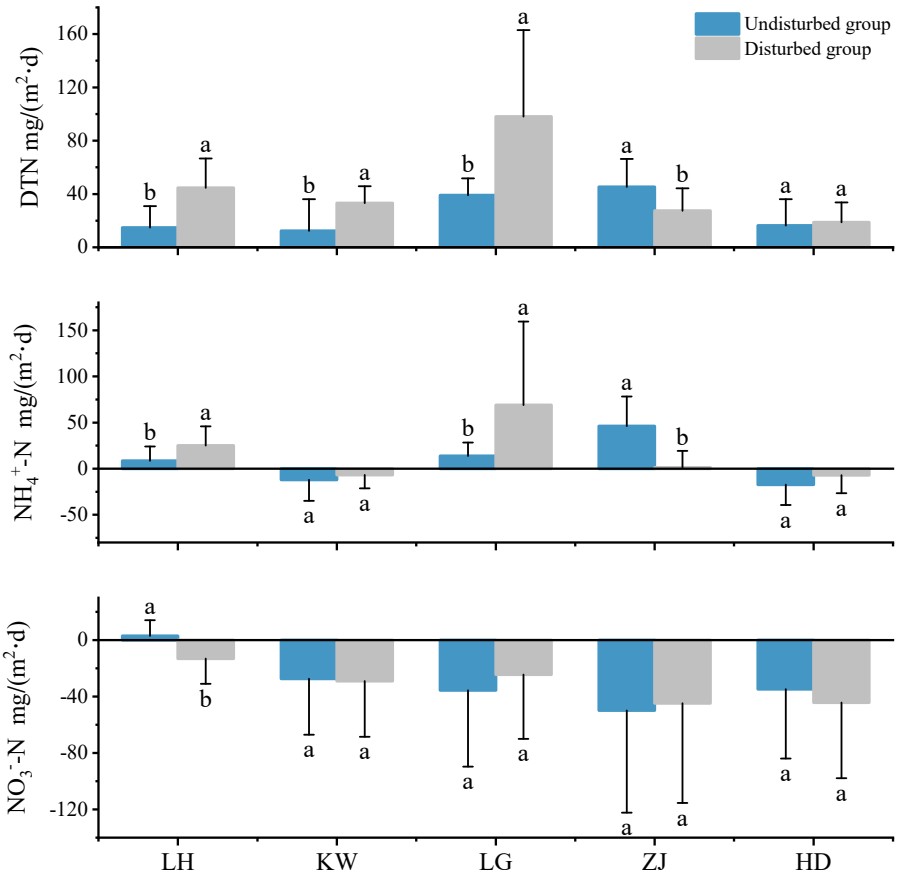

**Figure 5.** Diffusion fluxes of dissolved total nitrogen (DTN), nitrate ($NO_3^-$-N), and ammonia ($NH_4^+$-N) at different sampling sites during the experiment. LH, kW, LG, ZJ, and HD represent sampling sites. The labels on the error bar represent the results of the significance test, with identical letters representing no significant difference and different letters representing significant difference.

For shallow lakes, the disturbance was an important factor affecting the exchange of nitrogen between the overlying water and the sediment [29]. Qian et al. compared the exchange characteristics of nitrogen during undisturbed and vibrational resuspension of sediments, where the concentration of three states of nitrogen ($NO_3^-$-N, $NH_4^+$-N, and DTN) in water increased significantly during continuous (700 min) resuspension of sediments [30]. In fact, in a much longer experiment time, the effect of disturbed on DTN decreased with time. As shown in Figure 4, the flow disturbed accelerated the load accumulation of DTN and $NH_4^+$-N and reduced the release of $NO_3^-$-N. The effect of flow disturbed on the increase of DTN flux was temporary and only lasted for approximately ten days (Figure 4). However, in the relatively long term, the flow disturbed reduced the load accumulation of DTN (Figure 4). Therefore, the disturbed flow could reduce the load accumulation of DTN in the overlying water, and further research would be needed to investigate whether it can reduce the eutrophication in lakes.

*3.3. The Comparison of Diffusion Flux between Sediment Incubation and Model Calculation*

A comparison was made for the nitrogen and phosphorus fluxes of Longjing Lake between the results calculated by the model in Pan's study and the results by incubation in this study. The results showed differences in ammonia fluxes between both model

calculations and incubations (Table 1). Although there was no significant difference in LH and KW ($p > 0.05$), the direction of nutrient release was reversed in KW. There was also an order of magnitude difference in LH. Therefore, there was large variability in the results obtained by the two methods. When comparing the flux results of SRP, it could be found that the test results of ZJ showed significant differences because of the direction of nutrient release. An order of magnitude difference can be visually observed between the data on $NH_4^+$-N and SRP diffusion fluxes obtained from model calculations and sediment incubation. In terms of SPR release, sediment incubation results were consistently higher than model calculations, except for the LG interference group. Therefore, the modeling results based on Fick's law might underestimate the release potential of DTN and SRP.

**Table 1.** Comparing the results of diffusion flux based on model calculation and incubation.

| Sites | Flux of $NH_4^+$-N (mg/d·m$^2$) | | Flux of DTN (mg/d·m$^2$) | | Flux of SRP (mg/d·m$^2$) | | Flux of DTP (mg/d·m$^2$) | |
|---|---|---|---|---|---|---|---|---|
| | Model Calculation [a] | Incubation [b/c] | Model Calculation [a] | Incubation [b/c] | Model Calculation [a] | Incubation [b/c] | Model Calculation [a] | Incubation [b/c] |
| LH | 28.3 | 8.46/25.0 | - | 14.8/44.6 | 3.94 ** | 19.4 **/8.59 | - | 18.4/8.20 |
| KW | 2.03 | −12.0/−6.86 | - | 12.4/33.1 | −1.93 ** | 9.59/12.5 ** | - | 2.34/12.5 |
| LG | 47.2 * | 13.9/69.0 * | - | 39.1/98.0 | 6.13 * | 12.2 */4.67 | - | 11.0/6.71 |
| ZJ | 30.1 * | 46.1/1.14 * | - | 45.4/27.4 | −2.50 ** | 9.41/8.06 ** | - | 5.96/6.31 |
| HD | 14.6 | −17.2 */−7.10 * | - | 16.3/18.7 | 0.839 * | 8.59/8.52 * | - | 4.75/6.41 |

[a], the results of the model calculation were cited from Pan's study (Pan et al., 2014); [b], the results of the undisturbed group; [c], the results of the disturbed group; *, the results of the significant difference test (T test of the undisturbed group and the disturbed group relative to the model calculation, * $p < 0.05$, ** $p < 0.01$).

The variation between the two methods might also be caused by the biological or physical factors affecting nutrient release in the actual environment, such as biological activities (i.e., benthic organisms) and biochemical transformation factors (i.e., feeding and excretion) [31]. Eek et al. summarized nine factors that can have a significant impact on fluxes, including molecular diffusion, colloidal diffusion, bio-irrigation, bioturbation, particle resuspension, wave pumping, submerged groundwater discharge, and advection [32]. Considering these factors in the lacustrine sediments, the fluxes obtained in incubation were the summary of all the contributions. However, the model only considered molecular diffusion because it was the only way that could be quantified mathematically. Thus, the fluxes calculated may be underestimated. Our results show that the model calculation was orders of magnitude lower than that obtained from the incubations (e.g., KW and HD). On the other hand, there are also some disadvantages to sediment incubations. For example, the sample collection process always tends to disrupt the concentration, distribution, and composition of the analytes in lacustrine sediments, whereas the porewater model avoids this process because the sampling process is nondestructive. It is a simple and effective method for characterizing the diffusion fluxes of nutrients, showing advantages in studies with small time scales.

## 4. Conclusions

Based on the comparative study of the two methods, the following conclusions can be obtained: (1) The fluxes obtained by the two methods have significant differences. Sediment incubation is a combination of multiple influencing factors to obtain fluxes, while the porewater model is based on molecular diffusion as the theoretical basis for obtaining fluxes. (2) In the management of autochthonous release from lakes, sediment incubation is recommended as a research tool when the main objective is to remove pollution, while the porewater model is recommended as a research tool when the main objective is to control pollution. (3) When assessing the diffusive flux of nitrogen, it is recommended to choose the stable form of total dissolved nitrogen to discuss the flux results.

**Supplementary Materials:** The following is available online at https://www.mdpi.com/article/10.3390/w14132015/s1, Figure S1 Diagram of laboratory simulation experiment apparatus; Figure S2 The accumulation of Phosphorus concentration; Figure S3 The accumulation of Nitrogen concentration.

**Author Contributions:** All authors contributed to the study conception and design. Material preparation, data collection and analysis were performed by C.D., Y.P., W.T., Q.Y. and H.Z. The first draft of the manuscript was written by C.D. and all authors commented on previous versions of the manuscript. All authors have read and agreed to the published version of the manuscript.

**Funding:** The research was supported by the National Natural Science Foundation of China (No. 41877471, and 41877368), the special fund from the State Key Joint Laboratory of Environment Simulation and Pollution Control (20L03ESPC, and 21Z02ESPCR).

**Institutional Review Board Statement:** Not applicable.

**Informed Consent Statement:** Not applicable.

**Data Availability Statement:** Not applicable.

**Acknowledgments:** The research was supported by the National Natural Science Foundation of China (No. 41877471, and 41877368), the special fund from the State Key Joint Laboratory of Environment Simulation and Pollution Control (20L03ESPC, and 21Z02ESPCR). Special acknowledgement to the reviewers and editors for their constructive comments.

**Conflicts of Interest:** The authors declare no conflict of interest.

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
