# Peer review of "A Comparison Study of the Nutrient Fluxes in a Newly Impounded Riverine Lake (Longjing Lake): Model Calculation and Sediment Incubation"

_water, doi:10.3390/w14132015_

Round 1

Reviewer 1 Report

General comment:

Experiments results should be assessed using statistic measures, not only visually on graphs.

Additionally, my request to rewrite the terms 'we collected', 'our objectives', and 'our sediments' into impersonal forms was not considered. Entire text still needs to be edited. I recommend editing the text by a native speaker.

Specific remarks:

Abstract:

The abstract is legible and contains the relevant parts, but needs improvement, especially the final fragment.

line 14: Formulation 'and / or' is not proper for a research paper.

lines 19-21: The same sentence twice.

line 26: The capital letter after dot missing.

lines 26-27: I don’t understand the sentence. Please edit.

lines 27-29: This sentence is even more incomprehensible. Please do not use an expression 'one should'. It does not sound good.

lines 35-38: The sentence has a incorrect syntax.

Line 37: Please revise if the algal bloom is a limiting factor for primary production as it is a primary production.

line 42: What does 'the extend of production of key factors' mean?

Study area: There is still nothing about the conditions and climate.

Sediment sampling: Only written where downloaded, nothing about the method, amount of sampling material, etc.

lines 89-90: The whole sentence put in brackets do not look correct.

lines 90-92: The sentence in incomprehensible, with doubled expressions.

Lines 104-109: The equation is unclear as well as its description.

Line 121: The chapter should not content only a equation.

Results and Discussion: The chapter starts with discussion without detailed results description.

lines 145-146: The sentence is incomprehensible.

line 149: Please explain what Authors mean to 'higher quality of DTP / SRP'.

lines 132-154: The fragment is about slopes comparison. On what basis, what statistics this comparison was made?

line180: 'nitrite and nitrite'

line 183: The significance of the change should be proved by statistics measures.

lines 186-187: The sentence is unclear.

Lines 189-191: Please edit the sentence, doubled expressions.

 lines 214-218: The sentence is too long and unclear.

Conclusions: Conclusions are indistinct and it is unclear what is a significance of research and an input to scholar society.

Author Response

We have uploaded the responses to the attachment.

Reviewer 2 Report

In this study, sediment samples were collected from a China’s lake, then the fluxes were estimated by model calculation and laboratory incubation. The results show that there is an order of magnitude difference in the fluxes measured by these two methods. The results show that there is an order of magnitude difference in the fluxes measured by these two methods. The mean values of ammonia nitrogen (NH4+-N) and soluble reactive phosphate (SRP) obtained from the model calculations were 24.4 and 1.30, respectively. The mean values of NH4+-N and SRP obtained for the sediment incubation static group were -0.864 and 11.8, respectively, and for the disturbance group -21.7 and 8.47, respectively. The results obtained by incubation were higher than that of model calculation, indicating the underestimation of the flux by models. total nitrogen flux was a better parameter compared to either ammonia or nitrate. Thus, one should be cautious to choose methods and parameters for the flux estimations of nutrients for lakes or reservoirs.

The topic is interesting but there are some points to be addressed. In particular, the aim of the analysis should be evidenced in. the text. The conclusions should be improved with the weaknesses of the analysis and the insights for future research.

Author Response

(The authors gave the same response as above.)
